# Bacterial Cellulose and ECM Hydrogels: An Innovative Approach for Cardiovascular Regenerative Medicine

**DOI:** 10.3390/ijms23073955

**Published:** 2022-04-02

**Authors:** Izabela Gabriela Rodrigues da Silva, Bruna Tássia dos Santos Pantoja, Gustavo Henrique Doná Rodrigues Almeida, Ana Claudia Oliveira Carreira, Maria Angélica Miglino

**Affiliations:** 1Department of Surgery, School of Veterinary Medicine and Animal Science, University of São Paulo, São Paulo 05508-270, Brazil; izabelarodrigues@usp.br (I.G.R.d.S.); bruna.pantoja@usp.br (B.T.d.S.P.); henrique.gustavo1436@gmail.com (G.H.D.R.A.); ancoc@iq.usp.br (A.C.O.C.); 2NUCEL-Cell and Molecular Therapy Center, School of Medicine, Sao Paulo University, Sao Paulo 05508-270, Brazil

**Keywords:** extracellular matrix, biomaterials, tissue engineering, heart regeneration

## Abstract

Cardiovascular diseases are considered the leading cause of death in the world, accounting for approximately 85% of sudden death cases. In dogs and cats, sudden cardiac death occurs commonly, despite the scarcity of available pathophysiological and prevalence data. Conventional treatments are not able to treat injured myocardium. Despite advances in cardiac therapy in recent decades, transplantation remains the gold standard treatment for most heart diseases in humans. In veterinary medicine, therapy seeks to control clinical signs, delay the evolution of the disease and provide a better quality of life, although transplantation is the ideal treatment. Both human and veterinary medicine face major challenges regarding the transplantation process, although each area presents different realities. In this context, it is necessary to search for alternative methods that overcome the recovery deficiency of injured myocardial tissue. Application of biomaterials is one of the most innovative treatments for heart regeneration, involving the use of hydrogels from decellularized extracellular matrix, and their association with nanomaterials, such as alginate, chitosan, hyaluronic acid and gelatin. A promising material is bacterial cellulose hydrogel, due to its nanostructure and morphology being similar to collagen. Cellulose provides support and immobilization of cells, which can result in better cell adhesion, growth and proliferation, making it a safe and innovative material for cardiovascular repair.

## 1. Introduction

Cardiovascular diseases are considered the major cause of deaths around the world among non-transmissible pathologies. The number of deaths for these diseases increased from 12.1 million in 1990 to 18.6 million in 2019 [1]. According to the World Health Organization (WHO), myocardial ischemia represents 16% of all deaths globally [2]. Although myocardial ischemia is the main cause of death in humans, cardiomyopathies are the second most relevant [3,4]. With less information than for humans, heart diseases are described in the same way for domestic animals such as dogs and cats, which are widely affected by acute myocardial infarction and myocardial ischemia [5,6].

In tissue engineering, regenerative medicine uses extracellular matrixes (ECMs) as a natural model for bioactive modifications. Hydrogels are biopolymers widely used due to their three-dimensionality. Hydrogel production makes it possible for the application of ECM as a model for biomimetic scaffolds, which offers a temporary structure for cell delivery in a three-dimensional space [7,8].

Cellulose is the most abundant biopolymer in nature, a fibrous and water-insoluble compound, found in the cellular walls of plants. However, it also may be produced by some animals, fungi and bacteria. Because of the high presence of hydroxyls in the molecular structure of cellulose, it can be used for hydrogel production with several structures and properties to act as a platform for tissue engineering [8].

## 2. Cardiovascular Diseases: An Overview into Human and Veterinary Medicine

Cardiovascular diseases (CD) were recognized as a main world health concern in the last twenty years, despite a half-century of advances in the preventive medicine field [9]. The 2019 Global Burden of Disease (GBD) Study showed that the total number of cardiovascular disease cases nearly doubled from 271 million in 1990 to 523 million in 2019. After an injury, the myocardium muscle cells are replaced by fibrotic tissue, and then, during the remodeling process, the activated cardiac fibroblasts became myofibroblasts, promoting stiffness and fibrosis, which are associated with cardiac insufficiency and an unfavorable prognosis [10]. Despite decades of great advances and efforts to treat CD, the search for alternative therapies remains due to the pandemic context of cardiac insufficiency. The treatment for myocardial infarction is still a clinical challenge because of the limited myocardial regeneration [11].

A large-scale myocardial infarction determines that a patient may lose about one billion healthy cardiomyocytes. The ischemic area is commonly infiltrated with inflammatory cells, which are replaced by myofibroblasts [12,13,14]. In this context, the heart transplant continues to be the definitive treatment for cardiac insufficiency; however, the supply of donor organs is limited and the procedure may cause lifelong immunosuppression, hypertension, diabetes and renal insufficiency [15,16]. Although not considered definitive therapies, revascularization interventions or drug treatment in some cases can prevent the need for transplantation [14,17].

Although myocardial infarction is one of the leading causes of sudden death in humans, the second most important cause is cardiomyopathies, such as dilated cardiomyopathy (DCM), hypertrophic cardiomyopathy (HCM), and arrhythmogenic right ventricular cardiomyopathy (AVCD) as well as the “electric diseases” without structural abnormalities [3,4]. Interestingly, although myocardial infarction is not often reported in dogs and cats, other diseases are similarly described as a cause of sudden death in these companion animals [5,6]. As in humans, the most common structural heart diseases associated with sudden death are DCM and CAVD. “Electrical diseases” can also cause sudden death in dogs; however, there is considerably less information about these disorders when compared to humans. In veterinary medicine, acute myocardial infarction is a somewhat uncommon disease in companion animals; however, such species play an important role in acute and severe heart failure [18,19,20].

Dilated cardiomyopathy (DCM) can be defined as a structural disease in which left ventricular dilatation with systolic dysfunction occurs in the absence of conditions of abnormal loads or coronary disease capable of causing generalized systolic impairment. A distinction is made between “primary” DCM, for example often due to a genetically inherited defect, and CMD-like cardiac phenotypes that are “secondary”, for example, acquired from another cause [21]. In dogs, there is a racial predisposition to primary CMD, which suggests a genetic basis with familial transmission. Among the high-risk breeds are: Doberman pinscher, Irish Wolfdog, Newfoundland, Boxer, Great Dane, Saint Bernard, Portuguese Water Dog and German Shepherd. The prevalence of this disease is highly breed-dependent and ranges from 10% in Newfoundland to a cumulative prevalence of 58.8% in Doberman pinscher [22,23]. Doberman pinschers, Boxers and Great Danes are particularly prone to sudden cardiac death. Although actual data on disease prevalence are sparse, sudden cardiac death occurs in approximately one-third of preclinical Dobermans and in more than 30% of dogs with clinical signs, it appears to be much more common than in other dogs [20,24,25,26]. By correlating sudden death with CMD, two mutations in the *phospholamban* and *titin* genes, also identified in humans, were recently described in dogs with CMD.

Hypertrophic cardiomyopathy (HCM) is a structural disease characterized by ventricular hypertrophy, myocardial disorganization and fibrosis [27,28,29]. HCM causes diastolic dysfunction due to hypertrophy of the left ventricular free wall and interventricular septum, resulting in filling of the ventricles and atrial dilatation [27,30]. In dogs, primary HCM is extremely rare. When it occurs, it is through pathogenesis very similar to the HCM that occurs in humans and cats [20,31]. Ten to fifteen percent of cats are affected by HCM, and some studies report a prevalence of up to 30% in elderly cats older than 9 years [20,32,33,34]. During a 2-year retrospective follow-up period, about 4.7% of HCM cats had a sudden death, although the reported total mortality of 55.3% was much higher [35].

Arrhythmogenic right ventricular cardiomyopathy (AVCD) is characterized by atrophy and fibrofatty infiltration of the right ventricle. The pathophysiology of CAVD in dogs and cats is similar to the disease in humans. In Boxer dogs, the histopathological findings are very similar to humans, which is why these animals are an ideal spontaneous model for ARVC in humans [3,20,36,37]. The prevalence of the disease in humans in the general population has been estimated to be around one in 5000 [20,38]. In dogs, the exact prevalence is unknown. There is a great predilection for the breed, directly affecting Boxers and English Bulldogs, although they have already been described in Weimaraner and a Husky [3,36,37,39,40,41,42]. Sudden death was reported in 39% of the 23 Boxer dogs that were followed in clinical trials [3].

Conventional human therapeutic methods available in medicine include the use of coronary artery bypass, coronary reperfusion therapy, and fibrinolytic therapy, which alleviate acute symptoms, rather than providing repair and regeneration of damaged tissue [43]. Transplantation or a ventricular assist device (LVAD) [44] is the ultimate method of treatment for patients with heart failure. The main objective of the treatment of cardiomyopathies in veterinary medicine is to control clinical signs, delaying the evolution of disease, providing a better quality of life for patients and reducing mortality, since these are diseases that have no cure. The reality of organ and tissue transplants in veterinary medicine is completely different, and undeveloped compared to human medicine, due to several factors. There is no turnover of organs for transplants, as there is in human medicine, in which there is a specific program for the availability of these organs; the elaboration of ethical-legal legislation for transplants and transplanted, and the development of less invasive surgical techniques and the prognosis with negligible risk of rejection being necessary. The difficulty of performing the technique in veterinary medicine is intensified not only by the difficulty of finding compatible and available tissue and organs but also by the possibility of rejection [20,45]. Although pharmacological treatments of β-blockers and angiotensin-converting enzyme (ACE) inhibitors [46,47] are beneficial for patients with MI, these existing approaches require exploring new treatment methods aimed at regenerating the infarcted myocardium, as well as their implementation in clinical practice [48].

In turn, the heart has a limited regenerative capacity [49]. Although there are several medical and surgical therapies available, the body’s inability to regenerate the myocardium poses a major risk for patients with heart failure [50]. The use of stem cells is a promising approach for myocardial regeneration, and the concept of replacing cells lost in myocardial injury with new stem cell-derived cardiomyocytes attracts researchers. Several cell types are investigated for therapeutic purposes, from adult stem cells or progenitor cells to induced embryonic or pluripotent stem cells [51].

Cell therapies significantly captivate the field of cardiac regeneration; however, challenges include low cellular resistance after implantation and immune rejection, which are not easily resolved by conventional cell therapy methods. The tissue engineering specialty combines the use of biomaterials, cells and growth factors to manufacture or regenerate, for example, damaged myocardium. Regenerative medicine that uses cardiac tissue engineering techniques aims to improve patient’s survival and quality of life. Biomaterials can improve cell therapy by aiding cell survival, providing sufficient mechanical strength to house cells, which disintegrate as tissue develops, and their degradation products must not be toxic and must be eliminated from the body safely [51,52].

Cardiac tissue engineering, a technique associated with regenerative medicine, represents an effective approach to repair or regenerate damaged tissues and organs and restore their function. Over the past few years, the possibility of creating custom-made scaffolds with physicochemical and biomechanical characteristics, biomimetic devices based on synthetic or natural polymers, has been reflected in the increased interest in the area [53,54]. Cardiac scaffolds based on natural or synthetic biomaterials can mimic the environment of the extracellular matrix, releasing bioactive molecules. Thus, several types of injectable or implanted scaffolds are being proposed so far [55,56,57].

A biomaterial designed for cardiac tissue engineering needs to have fundamental properties to prevent myocardial dilation, avoid or delay scar formation or fibrosis, while favoring the integration and proliferation of cardiomyocytes [58]. At the same time, it needs to allow the interaction with all components of the myocardium (e.g., cardiomyocytes, endothelium, fibroblasts and perivascular cells), and the compatibility of metabolic by-products, the blood-material interaction, since it becomes a challenge when the exposure of the material to blood flow may result in thrombosis or embolism events. However, a good scaffold needs to have high biocompatibility and non-immunogenicity to avoid adverse effects during the healing process. In addition to having a degree of porosity in the range of 50 to 90%, to promote the diffusion of nutrients, oxygen and extracellular fluids through cellular networks, it must demonstrate mechanical properties that allow the mechanical resistance of the organ to be maintained until complete regeneration. Likewise, there should be a balance between the rigidity and flexibility of the organ to support repeated stretching cycles, in a way that does not limit the contractions and relaxation of the heart muscle, for example [59].

## 3. Tissue Engineering: Extracellular Matrix

Tissue engineering and regenerative medicine use extracellular matrix as a natural model for bioactive modifications. The production of hydrogels provides opportunities to use the natural extracellular matrix as a model for biomimetic scaffolds. In regenerative medicine, the main function of the scaffolds is to offer a temporary structure to cell delivery in a three-dimensional space [7,8].

The ECM is constituted of structural and regulatory proteins and polysaccharides and is generated and maintained by cells. A different ECM composes each organ. ECM coordinates cellular functions like proliferation, migration and differentiation, just as the matrix provides mechanical strength to tissue and organizes cells at specific sites. Proteins and glycans are two essential ECM components. Such proteins act as a scaffold and allow cell adhesion. In this way, the “cell-matrix” adhesion system mediates several physiological responses [60,61,62]. Cellular receptors bind to soluble and bound signals in the matrix environment; such receptor–ligand interactions trigger cascades of intracellular enzymatic reactions that regulate gene and protein expressions and determine the cell fate in a specific tissue [61]; likewise, the cell can emit a signal to build and degrade its microenvironment. Matrix characteristics are pertinent to tissue engineering, and in tissue engineering that seeks to replicate the composition and structure of the extracellular matrix. Therefore, the use of three-dimensional scaffolds, both natural and synthetic, can be produced to repair or restore damaged organs and tissues [60,63].

Like other natural polymers, scaffolds from decellularized tissues earned visibility in cardiac tissue engineering due to their ability to mimic the biophysical and topographic properties of native ECM [64]. The main sources of these decellularized scaffolds in this field are myocardium and pericardium [65]. Once the ECM is decellularized, it can be lyophilized, ground, enzymatically digested and transformed in a hydrogel [66]. Wainwright et al. [67] prepared a decellularized ECM from adult swine hearts to produce a suitable microenvironment for cardiomyocytes’ adhesion and proliferation. Liguori et al. [68] demonstrated that a swine cardiac ECM hydrogel can be loaded with trophic factors, which are secreted from the adipocyte-derived stromal cells, and can also be released in a sustainable way for several days.

## 4. Bacteria Cellulose: An Innovative Biomaterial

Bacterial cellulose was discovered two centuries ago, however, only in the last few decades, with the development of green chemistry and nanotechnologies, is it gaining space in the research community both in the academic and industrial fields. It is a versatile nanomaterial of commercial interest due to its natural purity, biodegradability, biocompatibility and non-cytotoxicity [69]. Since its discovery, significant research has focused on its production, manufacturing and new applications. Currently, bacterial cellulose has been widely used in drug delivery, tissue engineering, wound dressing, food and cosmetics [70,71].

Cellulose is the most abundant biopolymer on the planet, despite the vast biochemical and phylogenetic diversity of living beings. It is a fibrous, resistant substance insoluble in water, found in the protective cell walls of plants, mainly in stems, trunks and woody portions, being the main structural component of plants giving them mechanical and structural integrity. However, some animals (such as urochordates), fungi and some bacteria also produce it [72,73,74,75,76]. There are other routes of cellulose synthesis, such as chemosynthesis and enzymatic synthesis from glucose derivatives. Cellulose can be classified into two types according to the production origin. There is cellulose from plant biomass, which stands out as a source of raw material for the production of bio-based fuels, paper, packaging and biomedical applications [77,78]. Cellulose is derived from a variety of microorganisms such as fungi, algae (*Valonia ventricosae, Glaucocystis*), and bacterial strains belonging to the genera *Agrobacterium, Aerobacter, Achromobacter, Sarcina, Acetobacter, Rhizobium, Salmonella* and *Azotobacter* that produce acetic acid [79,80,81]. *Gluconacetobacter xylinus* (formerly known as *Acetobacter xylinus* and later *Komagataibacter xylinus*) can produce bacterial cellulose in greater amounts when compared to other species [82]. Such bacteria produce bacterial cellulose in a biosynthetic pathway, involving the secretion of polysaccharides formed while using carbon sources in the medium. Carbon sources such as glucose, sucrose, fructose and glycerin are often used in culture media to produce bacterial cellulose [83]. (Figure 1). 

The chemical structure of vegetable cellulose and bacterial cellulose are the same. However, bacterial cellulose has advantages such as high purity (as it is free of lignin and hemicellulose), high crystallinity (84–89%), high water retention (100 times its dry weight), good mechanical properties and 3D nanofibrous structure [84,85,86,87,88] These characteristics make BC a potential material for different applications [89].

Louis Pasteur initially defined bacterial cellulose as a moist, gelatinous skin-like substance produced by the fermentation of coconut water. Later, in 1886, Adrian Brown systematically reported bacterial cellulose as a gelatinous membrane formed on the surface of *Bacterium aceti* culture medium during acetic acid fermentation [90]. Soon after its discovery, such a membrane was called the “vinegar plant”; the producing microorganism was initially called *A. xylinum* [91], according to the International Code of Nomenclature of Bacteria, after *G. xylinus* [92,93]. This genus, *Komogataeibacter*, is currently the most studied species. It is a strictly aerobic gram-negative bacterium, present in fruits and vegetables in the process of decomposition. It is physiologically characterized by the production of acetic acid from ethanol, by the oxidation of acetate and lactate in carbon dioxide and water, being able to convert common carbon sources (glucose, glycerol, sucrose, fructose mannitol) at temperatures between 25 °C and 30 °C at a pH of 3–7.

There are different fermentation methods and such methods produce bacterial cellulose with different characteristics and applications; there are several methods to control fermentation and increase yield or obtain bacterial cellulose with different characteristics. Commonly, cellulose can be produced by static and agitated cultivation methods. To produce bacterial cellulose, bacteria use several carbon sources, such as glucose, fructose, mannose, glycerin, ethanol and pyruvate [94].

Under static conditions, bacteria need to float on the surface of the medium to obtain oxygen at the surface. Such bacteria produce cellulose at the interface of air and culture medium, similar to a film for a flotation mechanism, which allows the bacteria to stay in the air/liquid near the interface to get the oxygen needed for their metabolism. The film obtained forms a physical barrier protection against UV radiation, increases the ability to colonize other substrates and maintains its hygroscopic nature, enables moisture retention and prevents dehydration [83,89]. Bacterial cellulose produced under static conditions normally has high crystallinity and tensile strength [89]. Under agitation conditions, the bacteria form a fluffy, spherical or irregular cellulose in the medium; Unlike the static method, cultivation under agitation fills the culture medium with oxygen and enables faster cell growth. Cellulose obtained by stirring has a higher water retention capacity, lower Young’s modulus and crystallinity [95]. (Figure 2).

## 5. Cellulose Structure

Vegetable cellulose is formed as a lignocellulosic polymer, that is, its cellulose molecules are strongly linked to others, such as lignin and hemicellulose and several others. Any accessory molecules that follow cellulose have specific functionalities in plant physiology. Furthermore, the cellulose content in plants depends on natural sources. Cellulose has a high content of impurities, which requires several molecular adjustments for later application, such as in biomedicine. Furthermore, the purification and isolation of plant cellulose is an arduous process, which involves complex mechanical treatments followed by chemical or enzymatic pre-treatments [96]. The pulp purification processes on an industrial scale generate high costs and great environmental risks due to the degree of toxicity. On the other hand, bacterial cellulose is obtained in a highly pure form, and its purification process is simple, ecologically correct and low cost [97].

Bacterial cellulose is a biomaterial, which can be obtained in a pure form, consisting of glucose and water units. It has a 6-membered cyclic structure with reactive primary and secondary hydroxyl groups; wherein the β-D-glucopyranose ring, all -OH groups are free, playing an essential role for the intermolecular H bond between two adjacent chains. Unlike plant cellulose, bacterial cellulose has a completely crystalline core surrounded by a less crystalline zone interpolated by the amorphous form of cellulose, as well as an arrangement of fibers in a 3D lattice structure. Its fibers tend to self-assemble because of strong interactions and hydroxyl groups, such fibers constitute a network structure interconnected by intramolecular hydrogen bonds, forming sheets with high surface area and high porosity [90]. It contains no hemicellulose or lignin and only a small amount of carboline and carboxyl moieties [98]. The tensile strength of cellulose is between 200–300 MPa, and its Young’s modulus is up to 15–35 GPa [99]. Such mechanical properties are a direct consequence of the crystalline structures of nano and microfibrils. Furthermore, the association of high crystallinity, high content and water are responsible for the thermal stability of the biomaterial [100].

## 6. Bacteria Cellulose Properties

Bacterial cellulose has several properties, including porosity, mechanical properties, biocompatibility and biodegradability. Some studies in tissue engineering demonstrate that microporous and nanoporous scaffolds are suitable for cell growth [101]. In this way, the 3D porous structure, which allows better cell mobility, is a property of great importance in a biomaterial within tissue engineering, as this characteristic allows better mobility of cells or active agents in the transplant. Bacterial cellulose has membrane pores ranging from 100 to 300 nm, and the lack of macropores restricts the use of cellulose in some biomedical applications. Therefore, the association with gelatin, salt, sugar [102], polyethylene glycol [103], hydroxyapatite [104], sodium with calcium ions [105] is common to increase the porosity of the biomaterial.

The mechanical properties superior to those of vegetable cellulose are attributed to the cross-linked ultrafine fiber structure of bacterial cellulose [106]. Studies have shown that the force-deflection curves in single filaments present a value of 78 ± 17 GPa, as well as fibers aligned with macrofibers based on bacterial cellulose, presented a Young’s modulus of 16.4 GPa and the tensile strength of 248.6 MPa [107,108]. Wang et al. [109] prepared macrofibers based on bacterial cellulose through the drawing and wet twisting process. Such macrofibers showed deformation-dependent mechanical properties, that is, increasing the wet stretching stress, the tensile strength was increased to 826 MPa and the Young’s modulus was 65.7 GPa. Such mechanical properties can be improved with the association of nanomaterials, such as graphene [110,111], graphene oxide [112], silver nanowires [113]; for example. The incorporation of 8% graphene increased tensile strength by 68.8%, while incorporation with 5% graphene oxide improved the Young’s modulus of bacterial cellulose films by 10%, and the 30% graphene increased the tensile strength from ~15 MPa to ~185 MPa.

Biocompatibility can be defined as an adequate host response to the new material in each specific application and the absence of any toxic or allergic effects. Tissue compatibility is a basic and essential prerequisite for a new biomaterial. Such a property is possible due to the 3D nanofibrous network structure that allows cell penetration and proliferation [114]. Bacterial cellulose enables the growth of connective tissue cells, and it is a suitable material for the proliferation of different types of cells [115].

A material must be degraded in a timeframe that responds to the regeneration or healing process. There needs to be an adequate shelf life, no toxicity, and its mechanical properties must be biocompatible with the healing or regeneration process during degradation [102]. It is known that the cellulase enzyme degrades cellulose, and the absence of this enzyme in the human body makes the biomaterial non-biodegradable [116]. In this way, several works seek to increase its degradability, such as associating y-radiations, which degrade rapidly “in vivo” within 2 to 4 weeks [117].

## 7. Types of Hydrogels, Properties and their Applications

Hydrogels are networks of polymers, natural or synthetic, swelled by water that have several tissue-like properties and are widely explored and frequently used in tissue engineering [118,119,120]. 3D hydrophilic polymer networks are formed from molecular interactions between different functional groups present in base polymers, which swell with the absorption of biological fluids without undergoing any change in their underlying molecular structure. This feature allows hydrogels to act as a soft and elastic scaffold, being able to imitate tissue in a microenvironment [51]. Hydrogels interact with water or biological fluids by capillary force, penetration force, and hydration force, and these forces can cancel each other out [121], which interferes with the swelling of hydrogels driven by osmotic pressure and the Gibbs–Donnan effect. Hydrogels can simulate the natural microenvironment of cells, thus being one of the most common supports of tissue engineering [122]. The good uniformity and operability of hydrogels allow for expanded applications in various fields [123].

Hydrogels are important due to their in vivo swelling properties and mechanical strength, and their compatibility with biological tissues [124,125,126]. The application of hydrogels in cardiac therapy is observed as a means of thickening and stabilizing the myocardium via tissue volume, as well as for the administration of various therapies, such as cell and growth factors [127,128,129].

Mechanical properties are important for both pharmaceutical and biomedical fields. Such properties are critical for a hydrogel’s successful application as a drug delivery system. This allows its physical integrity to remain intact until cargo molecules are released at a predetermined rate for a predetermined time. The swelling properties, when the hydrogel absorbs water or aqueous fluids without dissolving, continue until there is an equilibrium between the water and the polymer; on the other hand, the degree of elasticity of the biomaterial from the polymer–polymer interactions makes it impossible for water to flow inside the hydrogel [126,130]. The swelling capacity of hydrogels comes from the presence of hydrophilic groups in polymeric chains, which determines their wide use in biomedical applications [131]. In tissue engineering and regenerative medicine, hydrogels need to be compatible and non-toxic. Hydrogel biocompatibility deals with its ability to affect an adequate host response in a given application. Hydrogels operate as reversible gels with magnetic, ionic, H-binding, or hydrophobic forces, which play an important role in network formation [132].

Hydrogels can be classified as natural, synthetic and mixed. Natural hydrogels are made up of collagen [133], gelatin [134], hyaluronic acid [135], fibrin [136], agarose, dextran, alginate [137], chitosan [138] and cellulose [8]. Natural hydrogels have similar structures to the natural extracellular matrix and have good biocompatibility and functionality, however, they have some deficiencies such as batch differences in structure, performance during preparation, potential immunogenicity and considerably poor mechanical properties that limit their applications [139,140,141,142].

Synthetic hydrogels are formed by polyethylene oxide, polyethylene glycol (PEG) [143,144], polyvinyl alcohol, polyacrylamide, *n*-isopropylacrylamide (PNIPAM) [145,146], among others. Its composites can be molecularly altered according to the type of hydrogel required for block structure, molecular weight, mechanical strength and biodegradability. However, synthetic hydrogels are cross-linked by free radical initiators and cross-linking agents; the use of these crosslinkers has disadvantages such as residual unreacted monomers and residual crosslinkers or initiators that can result in inflammation or cytotoxicity [123,147].

Mixed hydrogels are based on the joining of synthetic and natural polymers and are known as hybrid hydrogels. Such hydrogels can be formed by covalently coupling synthetic and natural polymers by chemical couplings or polymerization. The advantages of this method are that it does not require complex bioconjugation during the preparation of bioactive synthetic polymers, however, the use of natural polymers of animal origin can cause immunogenic reactions and infections [84,123].

In cardiac tissue engineering, hydrogels need several properties such as biocompatibility, degradability, low or absence of toxicity and immunogenicity. The application of only natural or only synthetic hydrogels limits the desirable properties of each one. Synthetic hydrogels have greater control over mechanical and chemical properties, which makes them stable and reproducible, while they do not have a natural site for cell adhesion and a lower degree of biocompatibility. On the other hand, natural hydrogels are biocompatible and their biological properties allow better applicability in vivo; however, they have fast degradation, long gelation periods, low mechanical properties and electrical conductivity, and few antioxidant properties. The association of natural and synthetic hydrogels presents itself as a promising approach, thus, hybrid hydrogels that have biochemical and biomechanical environments of native cardiac tissue are essential for successful cardiac tissue engineering approaches.

## 8. Hydrogel: Decellularized Extracellular Matrix and Cellulose

The characteristics of a suitable bioactive hydrogel scaffold need to be similar to the structure and biological properties of the extracellular matrix of natural tissue. Current bioactive polymer hydrogels are limited in simulating various biological functions and mechanical properties of the matrix. A decellularized matrix consists of a natural scaffold prepared from tissues by removing cellular components and retaining the 3D structure of tissues or organs and some components of natural fibers, such as collagen. The scaffold is biocompatible, non-immunogenic and biologically active. A hydrogel-based on the use of decellularized extracellular matrix retains several transforming growth factors, which can enhance cell growth, migration, proliferation, differentiation and angiogenesis; such interaction with cells enables the remodeling of tissue and organ structure and is crucial for the regeneration and functional repair of tissues and organs [123].

Hydrogels from decellularized extracellular matrix have several advantages, such as injectability, since the viscous fluid pre-gel can be injected and polymerized at physiological temperature to form a hydrogel that adapts to the shape of the defect site; having biological activity inherent to the natural matrix; not containing immunogenic cellular material; demonstrate adjustability of their mechanical properties, which can be controlled by concentration or crosslinking. The gelled decellularized matrix has a three-dimensional structure suitable for cell growth. In turn, hydrogels are modifiable and can support cells, therapeutics, drugs and other bioactive molecules. The machinability of hydrogels represented by 3D geometric molecular shapes can be characterized by 3D printing. Thus, the applicability of hydrogels encompasses both “in vivo” tests (in organs such as the heart, liver, lung, brain, colon, spinal cord) and “in vitro” tests (as a substrate for cell culture, biliary tree reconstruction, organoid culture, bioinks derived from the decellularized extracellular matrix) [123,148,149,150] (Figure 3).

The characteristics of scaffolds derived from decellularized extracellular matrix have gained attention in tissue engineering. In the table below, we list some studies that produced extracellular matrix hydrogels for cardiac tissue regeneration. (Table 1).

Cellulose-based hydrogels are used in various fields related to tissue engineering, such as bioactive cartilage implants; prototypes of blood vessels [161]; dressings [162]; surgical implants [163]; drug delivery [164]; artificial corneal grafts [8]; and dental implants [165]. Some BC-based products have already been commercialized, such as BioFill^®^, Bioprocess^®^, XCell^®^ and Dermafill^TM^, which are examples of bio-based membranes that have the main characteristics necessary for an ideal dressing [164]. BASYC^®^ is used for artificial blood vessels and Gegiflex^®^ is available for tissue engineering [165]. Bacterial nanocellulose (NCB) has enormous potential for use as a scaffold in tissue engineering, as bacterial cellulose is more effective than plant cellulose, which justifies the fact that bacterial cellulose is the first choice in medical and health applications for tissue engineering [8]. This biomaterial has promising characteristics due to the similarity of its nanostructure and morphology to collagen, which makes cellulose an option for use in supporting and immobilizing cells. The architecture of bacterial cellulose-based materials can be designed at different scales, from the nano to the macroscale, controlling the biomanufacturing process. BC fibers are solid and, when used in combination with other biocompatible materials, produce nanocomposites particularly suitable for use in human and veterinary medicine [166].

Although bacterial cellulose has several properties that are of great value for tissue engineering and for several biomedical applications, numerous approaches are applied to change its physical–chemical and functional properties, such as porosity, crystallinity, chemical structures and functions, to fully explore the potential of bacterial cellulose. Bacterial cellulose can undergo both in situ and ex situ modifications (Figure 3). The in situ modification describes the exogenous molecules addition to the culture medium during cellulose biosynthesis, while the ex-situ modification describes the materials inclusion after bacterial cellulose biosynthesis and purification [83]. Such approaches seek to modify bacterial cellulose in order to expand its advantageous characteristics and solve its disadvantages (Figure 4).

The new in situ properties interfere with the nanofibers crosslinking. The main objective of such modification are new characteristics in the matrix, changing its biophysical properties. The additives become part of the nanofibers, interacting with the –OH portions present in the bacterial cellulose chains and forming new hydrogen bonds. Chitosan, a polysaccharide derived from chitin, has biocompatibility, antibacterial and antifungal properties. The combination of bacterial cellulose, in a dressing, exhibited favorable antibacterial activities and no cytotoxicity [167]. Zhou et al. [168] demonstrated that their bacterial cellulose bandage associated with collagen I and hydroxypropyltrimethyl ammonium chloride chitosan exhibited excellent antibacterial activity, cytocompatibility and promoted the growth and proliferation of NIH3T3 cells and HUVECs cells. Silver nanoparticles and polydopamine incorporated into bacterial cellulose demonstrated antibacterial activity, increased cell viability, showed no cytotoxicity to fibroblast cells, granulation tissue formation, angiogenesis and re-epithelialization upon histopathological examination [169]. Several nanotubes, nanosheets were also incorporated into cellulose culture media. Park et al. [170] produced hybrid compounds of bacterial cellulose and carbon nanotubes that showed osteoconductivity and osteoinductivity. Likewise, Khalid et al. [171] demonstrated that the bandage, composed of bacterial cellulose and carbon nanotubes, acted as a mechanical and antibacterial barrier to fragile healing tissue, aided in moisture retention, reduced inflammation, and resulted in efficient wound healing. Graphene nanosheets were incorporated into the bacterial cellulose matrix, resulting in decreased crystallinity, improved mechanical and electrical properties. Luo et al. [172] produced a compound that exhibited high tensile strength with 93% improvement compared to pure bacterial cellulose film. In addition, the film also showed excellent flexibility with good conductivity. The association between nano zinc oxide and bacterial cellulose increased porosity and pore sizes, which increased water vapor permeability (an important factor for a bandage), it also showed antibacterial activity, good physical properties, non-cytotoxicity and good biocompatibility [173]. Although in situ modifications allow a uniform material distribution, the fermentation conditions of the biosynthesis process limit the incorporation of other materials.

Ex-situ modifications seek to alter the physicochemical and functional properties of the matrix after biosynthesis and purification of bacterial cellulose. The nanometric materials can be aggregated through diffusion to pass through the network pores. This type of modification can be divided into the chemical modification and composites development [174].

In the chemical modification process, bacterial cellulose is treated with several chemical reagents to modify its chemical structure and incorporate additional functionalities. The most common chemical modification is oxidation but there are also modifications by acetylation [175], benzoylation [176], succinylation [177] and phosphorylation [178].

Oxidation seeks to add new functional groups to cellulose. Oxidized cellulose is the most precious by-product of cellulose, and several chemical and physical properties of oxidized cellulose can be obtained under various oxidizing conditions (nature, temperature, pH and reaction duration) [179]. Many agents can be used, such as hydrogen peroxide, persulfates, permanganates, nitrogen dioxide, chlorine dioxide and phosphoric acids [180]. However, water-soluble 2, 2, 6, 6-tetramethylpiperidine-1-oxyl (TEMPO) is widely used to oxidize cellulose. Oxidized BC has been investigated for different applications such as adsorption of heavy metals, oil removal and various biomedical applications [181,182]. In etherification, the reaction is carried out in two steps; in the first step, cellulose is activated by treatment with an alkaline solution, followed by an etherification reaction with monochloroacetic acid or its sodium salt. Carboxymethylcellulose is one of the most important cellulose derivatives and can be used in pharmaceutical, cosmetic, food and biomedical areas [183]. Sulfation synthesizes cellulose in sulfuric acid in isopropyl alcohol or with SO3-pyridine complex in ionic liquids [184,185]. Cellulose sulfate has as its main characteristic, anticoagulating, antivirus and antibacterial properties [186]. Benzoylation treats bacterial cellulose with benzoyl citrate, adding to the material the potential for sensors, piezoelectric materials and optical properties [187]. Phosphorylation is developed for textiles and flame retardant materials, as it can induce the formation of calcium phosphate making the material suitable for biomedical applications [84].

Despite its advantages, cellulose has no antibacterial capacity and moderate mechanical properties. The development of composites aims to improve some properties that limit the application of bacterial cellulose in biomedical and tissue engineering. To improve mechanical and biological properties, researchers have incorporated different types of materials into bacterial cellulose, including polymers, carbon-based nanoparticles, metal/metal oxide nanoparticles, and other inorganic nanoparticles [188].

Bacterial cellulose fragments were immersed in the chitosan solution followed by lyophilization to produce a scaffold to aid in ovarian cancer diagnosis. The scaffold obtained showed better interaction with the cells compared to pure BC [189]. JU et al. [190] produced a bacterial cellulose film, in which the cellulose suspension and the polyvinyl alcohol solution were mixed, followed by the incorporation of chitosan in bulk form or nanoparticle form. The bulk form of chitosan increased the mechanical and elastic properties of the film, while the nanoparticle form showed higher antibacterial properties. The gelatin and hydroxyapatite incorporation in bacterial cellulose showed a composite with high mechanical properties, positive cell adhesion, proliferation and differentiation [191]. Yan et al. [192] achieved a scaffold with reduced porosity, high mechanical properties and great in vitro biocompatibility, by incorporating bacterial nano-cellulose into alginate and collagen. The incorporation of graphene, a carbon nanomaterial with a 2D structure, and carbon nanotubes add to bacterial cellulose better mechanical, electrical and thermal properties [84,193].

In summary, several biopolymers and biomaterials can be incorporated into bacterial cellulose to improve its properties, reducing its applicability limitations. The in situ and ex-situ modifications are methods that work on the incorporation of these materials homogeneously. Although the in situ modifications present several advantages of materials aggregation, the method is limited because some materials do not support the biosynthetic process. On the other hand, ex-situ modification expands the range of materials that can be incorporated; however, scientists still seek completely homogeneous incorporation in this process.

## 9. Bacterial Cellulose for Cardiac Tissue Regeneration

Understanding the environment nanoscale is essential to produce biomaterials that mimic the cellular microenvironment. The environment properties employ a total influence on cell adhesion, proliferation, maturation and differentiation, and consequently generate impacts on the function of a tissue. Cellulose is a very versatile material with its adaptable properties that allow its application in systems with different chemical and biophysical environments. Cellulose-based biomaterials provide important advantages over conventional synthetic materials, which demonstrates their promise of advancing scientific knowledge. The role of the extracellular matrix is established, and we know that it not only allows cellular attachment but also sends biochemical and biophysical clues to the nascent cells and tissues. Such data support studies on the application of scaffolds of decellularized tissues and organs in tissue engineering and regenerative medicine. The mimicry of natural conditions both in the tissue and in the ECM requires adequate adhesion and growth properties that maintain the tissue’s normal structure, and the results of biopolymers’ application involving celluloses mentioned above reveal successful results.

The bacterial cellulose use in cardiac tissue regeneration still needs more studies. In the literature, only one study was found that tested the cellulose membrane viability, acting as an adhesive, loaded with co-cultured cells. Simeoni et al. [194] produced a patch loaded with skeletal myoblasts and mesenchymal stem cells that was surgically inserted into the epicardial region of the left ventricle, where they found that the cellulose patch can protect the myocardium against the deleterious effects and pathological remodeling of the ischemic heart; this beneficial result was not obtained only with cell therapy. Other studies demonstrated the applicability of cellulose, modified cellulose and its composites. Only Simoeni et al. [194] describe the bacterial cellulose use itself. Chen et al. [195] developed a polyurethane/cellulose scaffold that presented greater mechanical strength and essential characteristics for the survival and function of cardiac cells with native anisotropy. As such, Entcheva et al. [196] tested the potential of cellulose acetate and reduced cellulose scaffolds for the growth of cardiomyocytes in vitro. They attested that the surface of these materials promoted cell growth, while increasing gap junctions, and electrical functionality. Such studies open doors to new possibilities for applications of bacterial cellulose, at the same time highlighting the potential of this biomaterial in cardiac tissue regeneration.

## 10. Conclusions and Perspectives

In summary, the pandemic nature of cardiovascular diseases in human medicine, and the high prevalence and incidence in companion animals, highlight the importance of searching for the best therapeutic method. Since there are great challenges in performing transplants in both areas, it is necessary to search for new therapies. Over the years, several techniques and therapeutic approaches have been proposed to improve the regeneration of a compromised myocardium after myocardial infarction. Cell therapy is one such approach, of great interest and widely investigated, as well as the production of scaffolds from hydrogels that allows the use of the natural extracellular matrix as a scaffold, providing a three-dimensional structure to support cell attachment and formation of cells. An interesting factor is the wide range of works in the areas of development and characterization of cellulose hydrogels, demonstrating that these hydrogels have the potential for application in tissue engineering.

Bacterial cellulose is a biopolymer that is synthesized by different species of bacteria, some fungi and algae, under appropriate cultural conditions. Nanofibers are produced as an extracellular matrix and are arranged in a three-dimensional network that has unique characteristics such as high purity, high crystallinity, microporosity, moldability, mechanical properties, absence of toxicity, high water retention capacity. Such characteristics make bacterial cellulose an emerging biopolymer with great potential for various biomedical and tissue engineering applications. Despite the advantages of bacterial cellulose, this biopolymer lacks antibacterial and antioxidant activities that limit some biomedical applications, as well as difficulties in handling, maintaining and storing cellulose hydrogels. The association of bacterial cellulose with other synthetic and/or natural biomaterials (for example, chitosan, graphene and graphene oxide) seeks to overcome some of the limitations of bacterial cellulose.

There is enormous progress in the use of bacterial cellulose, and there are already cellulose-based products on the market. Although there are several associations of bacterial cellulose for biomedical and tissue engineering applications, works that seek to combine bacterial cellulose with an extracellular matrix are scarce, opening up great research opportunities. Likewise, there are no studies that seek to associate bacterial cellulose with extracellular matrix aiming at cardiac tissue regeneration. Such facts broaden the perspectives of studies in the search for a scaffold that helps in the repair or regeneration of myocardial tissue, since the heart has an almost null regenerative capacity. Although more studies are needed on the proper development of bacterial cellulose hydrogels, they have important applications in tissue engineering due to their high biocompatibility and environment-friendly properties.

## Figures and Tables

**Figure 1 ijms-23-03955-f001:**
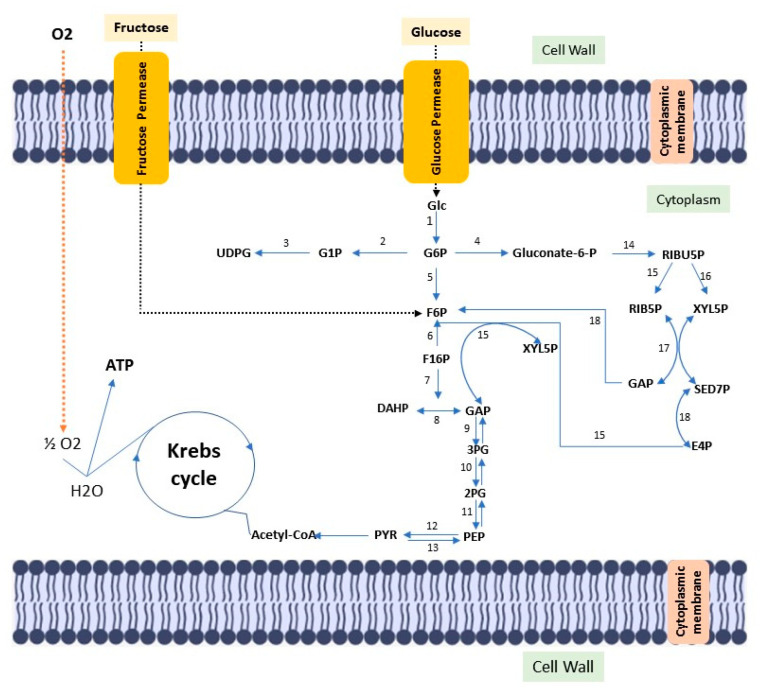
Diagram of the metabolic pathway, stimulated by fructose and glucose, for bacterial cellulose biosynthesis. Glc: Glucose; ATP glucokinase (1); GP6: Glucose 6-phosphate; Phosphoglucomutase (2); G1P: Glucose 1-phosphate; UTP–glucose-1-phosphate uridylyltransferase (3); UDGP: UDP-glucose; Glucose 6-phosphate dehydrogenase (4); Gluconate-6-p: Gluconate-6-phosphate; Phosphoglycoisomerase (5); F6P: fructose 6-phosphate; Fructokinase ATP (6); F16P: Fructose 1,6-bisphosphate; Aldolase (7); Triose phosphate isomerase (8); DHAP: Dihydroxyacetone phosphate; GAP: glyceraldehyde 3-phosphate; Glyceraldehyde 3-phosphate dehydrogenase (9); 3PG: 3-Phosphoglyceric acid; Phosphoglyceratomutase (10); 2PG: 2-Phosphoglyceric acid; Enolase (11); PEP: 2-phosphoenolpyruvate; Pyruvatokinase (12); Pyruvate diphosphate dikinase (13); PYR: Pyruvate; 6-phosphogluconate dehydrogenase (14); RIBU5P: Ribulose 5-phosphate; Phosphorribulose epimerase (15); Phosphorribulose isomerase (16); RIB5P: Ribose 5-phosphate; XYL5P: Xylulose 5-phosphate; Transacetolase (17); SED7P: sedoheptulose 7-phosphate; E4P: Erythrose 4-phosphate; GAP: glyceraldehyde 3-phosphate; Transaldolase (18).

**Figure 2 ijms-23-03955-f002:**
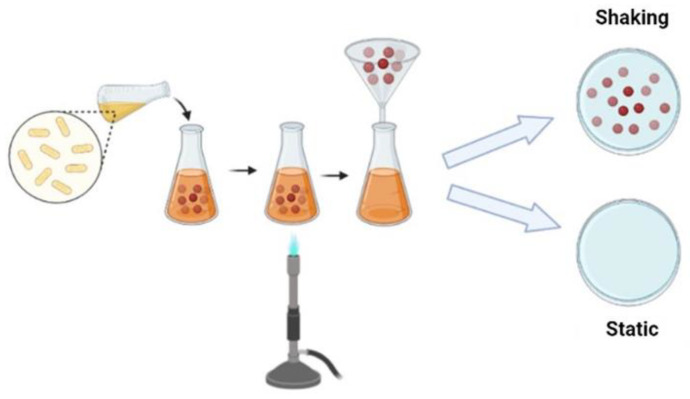
Schematic representation of the BC production strategy.

**Figure 3 ijms-23-03955-f003:**
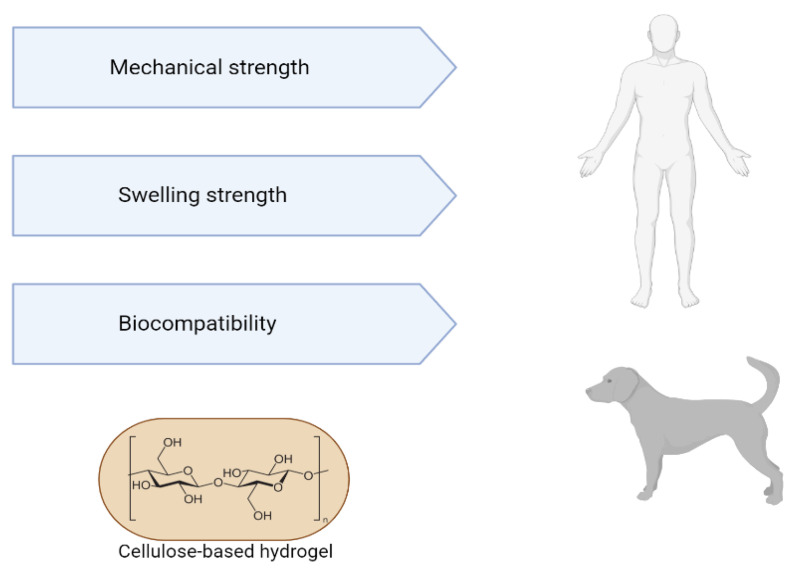
Advantages of using cellulose-based hydrogels for tissue engineering.

**Figure 4 ijms-23-03955-f004:**
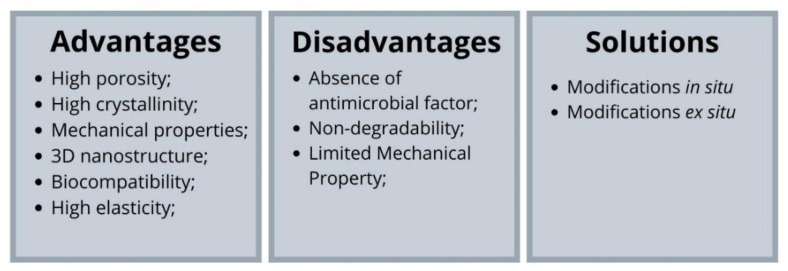
Advantages and disadvantages of bacterial cellulose and solutions to improve its properties.

**Table 1 ijms-23-03955-t001:** Types of extracellular matrix-derived hydrogels applied in vitro and/or in vivo cardiac repair.

ECM Origin (Organs and Specie)	Pure or Associated Hydrogel	in vitro or in vivo Assays	Specie (Assay)	Type of Repair	Concentration	Time of Treatment	Main Biological Findings	Reference
Porcine spleen	Pure	in vitro and in vivo	Mouse	Injectable hydrogel for induced myocardial infarction repair	SpGel + 1 × 10^5^ endothelial cells (iECs) and 2 × 10^5^ induced cardiomyocytes (iCMs)	4 weeks	Cardiomyocyte-specific marker proteins (α-actinin, cTnT and MLC2V); Cytoprotective effect; Encapsulation in SpGel increased the retention of cell grafts; It accelerated the cardiac function recovery, inhibited fibrosis and promoted the ischemic tissue revascularization.	[151]
Porcine myocardium and skeletal muscle	Pure	in vitro and in vivo	Pig and rat	Injectable hydrogel for induced myocardial infarction repair	-	3 months	ECM characterization of decellularized porcine skeletal and cardiac muscle, presenting a variety of characterization assays for ECM hydrogels of natural origin. These include evaluation of histology, DNA content, sulfated glycosaminoglycan (sGAG) content, mechanical properties (viscosity and storage and loss modules), protein content and nanoscale architecture.	[149]
Porcine myocardium	Pure	in vitro	Pig	Hydrogel for myocardial infarction repair	100 × ASC-ECM; 10 × ASC-MEC; 1 × ASC-ECM	5 days	Porcine decellularized cardiac ECM (dECM) hydrogels can be loaded with TFs secreted by human ASCs. The relative concentration of the trophic factor varies according to the concentration level of the hydrogel. Hydrogels can release trophic factors in a sustained manner, but each factor has its own kinetics.	[68]
Porcine myocardium	Pure	in vivo	Pig and rat	Injectable hydrogel for induced myocardial infarction repair	-	3 months	It demonstrates efficacy and feasibility in a clinically relevant porcine myocardial infarction model, where both pathophysiology and administration mimic what would be observed and performed in humans, as well as addressing important remaining safety issues. In addition to demonstrating the potential of an injectable myocardial matrix hydrogel to improve cardiac function, prevent negative LV remodeling, and increase cardiac muscle after MI in a porcine model.	[152]
Porcine ventricle	Pure		Pig and rat	Injectable hydrogel for induced myocardial infarction repair	-		It establishes a proof of concept for the clinical feasibility of the newly developed myocardial matrix as an injectable biomaterial for the treatment of myocardial infarction through a minimally invasive approach.	[153]
Porcine myocardium	Pure	in vitro and in vivo	Pig and mouse	Direct comparison on cell retention and therapeutic benefits of intramyocardial (IM) and intrapericardial (IPC) injection of adult stem cells in hydrogel. Induced myocardial infarction.	ECM + 2 × 10^5^ MSC	6 weeks	Better cell proliferation, less apoptosis and better vascular regeneration in the myocardium after intrapericardial delivery of MSCs. The CD63-RFP exosome tagging system showed that cardiac cells, including cardiomyocytes, took up MSC exosomes at higher rates using intrapericardial MSCs injection, compared to the results of intramyocardial injections, indicating more extensive paracrine activity of MSCs after intrapericardial injections.	[154]
Murine myocardium	Pure	in vitro and in vivo	Rat	The hydrogel effects on proliferation, cardiac differentiation and mutation were evaluated *systemically* in vitro. Next, the combination of BADSCs and temperature-sensitive ECM hydrogels was explored for cardiac regeneration and repair in MI models.	5 × 10^4^ BADSCs	4 weeks	Decellularized cardiac ECM can preserve intact native heart chamber geometry and most components of the extracellular matrix. Hydrogels had good bioactivity and regulated the behavior of stem cells in favor of myocardial repair, including cell survival, proliferation and cardiac regeneration.	[155]
Porcine myocardium	Pure	in vitro and in vivo	Rat	Characterizing the biochemical composition and structure of an injectable form of decellularized myocardial matrix, demonstrate its ability to form a gel in vivo, and assess its ability to promote a vasculature influx	Neonate rat cardiomyocytes (2 × 10^4^), RASMCs	11 days	The results of this study show the potential of an injectable form of myocardial matrix for use as an in-situ gelling support for myocardial tissue engineering.	[156]
Porcine myocardium	Pure; ECM + Hyaluronic Acid; ECM+ methacrylic anhydride and hyaluronic acid	in vitro and in vivo	Pig and mouse	Demonstrating that iPC injection can be an effective method to deliver multiple therapies to the heart.	-	-	Safety, efficacy and clinical feasibility of iPC injection of cardiac repair therapies. iPC injection could be developed as a new route for therapeutic administration.	[14]
Porcine myocardium	ECM and Polyurethane	in vivo	Rat	Heart patch. To assess the incorporation of a component of the cardiac extracellular matrix (cECM) and, secondly, to assess the impact of patch anisotropy on the pathological remodeling process initiated by myocardial infarction.	-	18 weeks	The most favorable remodeling response and better functional results would occur with the integration of the ECM into the patch by a change in the progression of several key effects of maladaptive remodeling after myocardial infarction, decreasing the global mechanical compliance of the LV, and nullifying the deterioration analyzed by echocardiography, mitigating scar formation and thinning of the LV wall and promoting angiogenesis.	[157]
Murine myocardium	ECM and Fibrin	in vitro and in vivo	Rat	Injectable hydrogel for induced myocardial infarction repair	340 μg mL ^−1^	21 days	ECM-Fibrin has adjustable composition and elastic modules that mimic the properties of developing and mature myocardium. The age of cardiac ECM development and the stiffness of the scaffolds affected cardiovascular gene expression and the formation of the c-kit+ CPC network in pediatric patients. The increase in the Young’s modulus of the scaffolds significantly inhibited the formation of the cellular network, suggesting different clues for differentiating pediatric c-kit+ CPC versus maturation.	[158]
Murine myocardium	ECM and inductive cocktail (oxytocin, ascorbic acid, vitamin E, beta-mercaptoethanol)	in vitro	Rat	To investigate the cECM effect on human adipose tissue-derived stem cells (hADSCs) differentiation into cardiomyocytes using cECM hydrogel in combination with a cardiac inductive cocktail.	2 × 10^5^ GFP-MSC	3 weeks	The cECM hydrogel alone can increase the proliferation of hADSCs and induce them to differentiate into cardiomyocyte-like cells. cECM was combined with an oxytocin-inducing compound, beta-mercaptoethanol, vitamin E and ascorbic acid. The gene expression of important early transcription factors (GATA4, NKx2.5, HAND1, HAND2), as well as structural genes and proteins connexin 43, cTnI, βMHC), increased considerably.	[159]
Murine myocardium	ECM and single wall carbon nanotubes	in vitro	Rat	Facilitate the development of cardiac seeding cell lineage in vitro and in vivo.	Group 1: intramyocardial injection of 100 µL of PBS; Group 2 treated with HH: intramyocardial injection of 100 µL of HH solution; Group 3 treated with BADSC: intramyocardial injection of 5 × 10^6^ BADSC in 100 µL of PBS; Group 4 treated with HH + BADSC: intramyocardial injection of 5 × 10^6^ BADSC in 100 µL of HH solution.	-	Modification of single-walled carbon nanotubes can improve bioactivity for building heart tissue resulting in a hybrid hydrogel that can be used as scaffolding for building heart tissue and injectable carriers for stem cell delivery. Hydrogel-associated nanotubes enhance the integrin-dependent niche through interaction with ECM proteins that will activate the integrin-related pathway and thus promote the development of primary and stem cell-derived cardiac cells towards functional tissues.	[160]

## Data Availability

Not applicable.

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
