# Peer review of "Bacterial Cellulose and ECM Hydrogels: An Innovative Approach for Cardiovascular Regenerative Medicine"

_ijms, 2022, doi:10.3390/ijms23073955_

Round 1
Reviewer 1 Report
The aim of the authors was to present a review devoted to Bacterial cellulose and ECM hydrogels: an innovative approach for cardiovascular regenerative medicine. Writing a review it is a big work which involves bringing new and important data related to the subject-matter addressed. Unfortunately it is not the case! The paper does not present any significant progress in the field. Additionally, the level of scientific depth is low. I got only a description of cardiovascular diseases, bacterial cellulose and hydrogels. Very little about the innovative approach for cardiovascular regenerative medicine.
As a whole, the paper shows a collection of data available in the literature with very little explanations/discussions/comments.
Author Response
Dear Editor and Reviewers,
We would like to thank you for your comments on our manuscript. All the comments were extremely valuable for the improvement of our article. Regarding this, we hereby inform you of the changes made to this manuscript, following the order of the questions raised.
Revisor 1:
The aim of the authors was to present a review devoted to Bacterial cellulose and ECM hydrogels: an innovative approach for cardiovascular regenerative medicine. Writing a review it is a big work which involves bringing new and important data related to the subject-matter addressed. Unfortunately it is not the case! The paper does not present any significant progress in the field. Additionally, the level of scientific depth is low. I got only a description of cardiovascular diseases, bacterial cellulose and hydrogels. Very little about the innovative approach for cardiovascular regenerative medicine.
As a whole, the paper shows a collection of data available in the literature with very little explanations/discussions/comments.
Reply: Thank you for your observations and suggestions and we agree with the reviewer. We communicate that each point noted by the reviewer has been redone. The structure of the manuscript was adequate, as well as discussions and comments were made.
Reviewer 2 Report
I have reviewed a manuscript entitled “Bacterial cellulose and ECM hydrogels: an innovative approach for cardiovascular regenerative medicine“. The manuscript tries to introduce bacterial cellulose hydrogel, as an ideal material for cardiovascular repair for human and pets. Overall, I found this manuscript interesting, however, I think it needs further modification before publication. My comments are below:
- Although the authors mentioned three figures in the main text, I could not find the figures in the attached files. Please include the figures in the main manuscript.
- I think it would be beneficial if you could add a table summarizing the published papers in this area.
- Some sections in the main text are too generic and can be removed without damaging the whole structure such as “Tissue Engineering: Extracellular Matrix”
- I think it would be better to highlight the disadvantages and advantages of bacterial cellulose in a table and provide potential solutions to overcome the associated challenges.
Author Response
Dear Editor and Reviewers,
We would like to thank you for your comments on our manuscript. All the comments were extremely valuable for the improvement of our article. Regarding this, we hereby inform you of the changes made to this manuscript, following the order of the questions raised.
Revisor 2:
I have reviewed a manuscript entitled “Bacterial cellulose and ECM hydrogels: an innovative approach for cardiovascular regenerative medicine“. The manuscript tries to introduce bacterial cellulose hydrogel, as an ideal material for cardiovascular repair for humans and pets. Overall, I found this manuscript interesting, however, I think it needs further modification before publication. My comments are below:
- Although the authors mentioned three figures in the main text, I could not find the figures in the attached files. Please include the figures in the main manuscript.
Reply: Thanks for the suggestion we agree with the reviewer and all figures were included in the main text.
- I think it would be beneficial if you could add a table summarizing the published papers in this area.
Reply: Thank you for the suggestion and we f agree with the reviewer. The suggestion was accepted and we inserted the table, including several articles that produced and applied extracellular matrix hydrogels for cardiac tissue regeneration.
- Some sections in the main text are too generic and can be removed without damaging the whole structure such as “Tissue Engineering: Extracellular Matrix”
Reply: Thanks to the reviewer for the suggestion and we agree with the comment. We modified the sections as suggested, such as “Tissue Engineering: Extracellular Matrix” to “The Role of Extracellular Matrix in Tissue Engineering”
- I think it would be better to highlight the disadvantages and advantages of bacterial cellulose in a table and provide potential solutions to overcome the associated challenges.
Reply: Thanks to the reviewer for the suggestion and we agree with the comment made. As suggested, we highlight the advantages and disadvantages of bacterial cellulose in figure 3 and describe examples from literature in the text.
Round 2
Reviewer 1 Report
The manuscript entitle “Bacterial cellulose and ECM hydrogels: an innovative approach for cardiovascular regenerative medicine” was revised and the version 2 it is an improved form.
I have few suggestions:
-Figure 1 it is or not updated from….and if it is the case the author should be specified
- Figure 2-too simple! It is necessary?
- A final English spelling and grammar of the text!
In conclusion, the manuscript could be published in the presented revised form.
Reviewer 2 Report
The authors have addressed the comments properly. I think the current version is acceptable for publication.